# Modeling microbial cross-feeding at intermediate scale portrays community dynamics and species coexistence

**Chen Liao** [1], **Tong Wang** [2,3], **Sergei Maslov** [3,4], **Joao B. Xavier** [1]*

**1** Program for Computational and Systems Biology, Memorial Sloan-Kettering Cancer Center, New York, New York, United States of America, **2** Department of Physics, University of Illinois at Urbana-Champaign, Illinois, United States of America, **3** Carl R. Woese Institute for Genomic Biology, University of Illinois at Urbana-Champaign, Illinois, United States of America, **4** Department of Bioengineering, University of Illinois at Urbana-Champaign, Illinois, United States of America

* XavierJ@mskcc.org

**Data Availability Statement:** All relevant data are within the manuscript and its Supporting Information files.

## Abstract

Social interaction between microbes can be described at many levels of details: from the biochemistry of cell-cell interactions to the ecological dynamics of populations. Choosing an appropriate level to model microbial communities without losing generality remains a challenge. Here we show that modeling cross-feeding interactions at an intermediate level between genome-scale metabolic models of individual species and consumer-resource models of ecosystems is suitable to experimental data. We applied our modeling framework to three published examples of multi-strain *Escherichia coli* communities with increasing complexity: uni-, bi-, and multi-directional cross-feeding of either substitutable metabolic byproducts or essential nutrients. The intermediate-scale model accurately fit empirical data and quantified metabolic exchange rates that are hard to measure experimentally, even for a complex community of 14 amino acid auxotrophies. By studying the conditions of species coexistence, the ecological outcomes of cross-feeding interactions, and each community's robustness to perturbations, we extracted new quantitative insights from these three published experimental datasets. Our analysis provides a foundation to quantify cross-feeding interactions from experimental data, and highlights the importance of metabolic exchanges in the dynamics and stability of microbial communities.

## Author summary

The behavior of microbial communities such as the human microbiome is hard to predict by its species composition alone. Our efforts to engineer microbiomes—for example to improve human health—would benefit from mathematical models that accurately describe how microbes exchange metabolites with each other and how their environment shapes these exchanges. But what is an appropriate level of details for those models? We propose an intermediate level to model metabolic exchanges between microbes. We show that these models can accurately describe population dynamics in three laboratory

**Funding:** This work (CL) was supported by NIH (National Institutes of Health, https://www.nih.gov) grants U01 AI124275 and R01 AI137269-01 to JBX. The funders had no role in study design, data collection and analysis, decision to publish, or preparation of the manuscript.

**Competing interests:** The authors have declared that no competing interests exist.

communities and predicts their stability in response to perturbations such as changes in the nutrients available in the medium that they grow on. Our work suggests that a highly detailed metabolic network model is unnecessary for extracting ecological insights from experimental data and improves mathematical models so that one day we may be able to predict the behavior of real-world communities such as the human microbiome.

## Introduction

Most microorganisms that affect the environments we live in [1] and that impact our health [2] do not live in isolation: they live in complex communities where they interact with other strains and species. The past decade has seen a surge of scientific interest in microbial communities, such as the human microbiome, but most studies remain limited to cataloguing community composition [3]. Our mechanistic understanding of how biochemical processes occurring inside individual microbial cells command interaction between cells, and lead to the emergent properties of multi-species communities remains limited [4].

Microorganisms consume, transform and secrete many kinds of chemicals, including nutrients, metabolic wastes, extracellular enzymes, antibiotics and cell-cell signaling molecules such as quorum sensing autoinducers [5–8]. The chemicals produced by one microbe can impact the behaviors of others by promoting or inhibiting their growth [9], creating multi-directional feedbacks that can benefit or harm the partners involved [10,11].

If a community is well-characterized and given sufficient data on population dynamics, it should be possible to parameterize the processes involved in microbe-microbe interactions by fitting mathematical models [12]. Any model can potentially yield insights [13], but the complexity of most models so far has been either too high for parameterization [14], or too low to shed light on cellular mechanisms [15]. Microbial processes may be modelled across a range of details: At the low end of the spectrum we have population dynamic models such as generalized Lotka-Volterra (gLV) [16] and Consumer-Resource (C-R) models [17], which treat each organism as a 'black-box'. For example, C-R models assume a linear or Monod dependence of microbial growth on resource uptake kinetics. At the high end of the spectrum, we have detailed single-cell models such as dynamic flux balance analysis (dFBA) [18] and agent-based models [19] that have too many parameters to be parameterizable by experimental data. For example, the linear equations for fluxes obtained from quasi-steady-state assumption of dFBA are underdetermined. What is an appropriate level of detail to model and constrain microbial processes using data, to produce accurate predictions as well as new mechanistic insights?

Here we propose a generalizable framework that couples classical ecological models of population and resource dynamics with coarse-grained intra-species metabolic networks. We show that modeling communities at this intermediate scale can accurately quantify metabolic processes from population dynamics data acquired in the laboratory. We demonstrate the approach on three evolved/engineered communities of *Escherichia coli* (*E. coli*) strains with increasing levels of complexity: (1) unilateral acetate-mediated cross-feeding [20], (2) bilateral amino-acid-mediated cross-feeding between leucine and lysine auxotrophies [21], and (3) multilateral amino-acid-mediated cross-feeding between 14 distinct amino acid autotrophies [22]. The parameterized models report inferred leakage fractions of metabolic byproducts that are difficult to measure directly by experiments, reveal how resource supply and partitioning alter the coexistence and ecological relationships between cross-feeders, and predict the limits of community robustness against external perturbations.

## Results

### Modeling microbial metabolic processes at an intermediate level

Inspired by the classical MacArthur's CR models [23] and many follow-ups [17,24–26], we propose to integrate CR models with a coarse-grained yet mechanistic description of cell metabolism. Metabolic reactions can be broadly classified as catabolic and anabolic, where catabolic reactions break down complex substrates from culture media into smaller metabolic intermediates that can be used to build up biomass components by anabolic reactions. A minimal representation of cell metabolism is a three-layer network composed of growth substrates at the top, building block metabolites in the middle, and biomass at the bottom (Fig 1). The growth substrates can be either substitutable (e.g., glucose and acetate) or non-substitutable (e.g., glucose and ammonium); however, in our model we consider only the non-substitutable building blocks for cell growth. In fact, substitutable metabolites can be mathematically lumped into complementary functional groups that together make a non-substitutable group when coarse-graining metabolic network. Despite its simplicity, this model is flexible enough to describe the transformation of resources into other resources, non-consumable chemicals and biomass, regardless of the specific reactions involved.

Based on these assumptions, we developed a dynamic modeling framework that contains eight kinds of biochemical reactions describing resource uptake, transformation, secretion, utilization, and degradation (Fig 1, S1 Text). Briefly, substrates available in the growth media can be imported into cells. A certain fraction of the imported substrates is then broken down

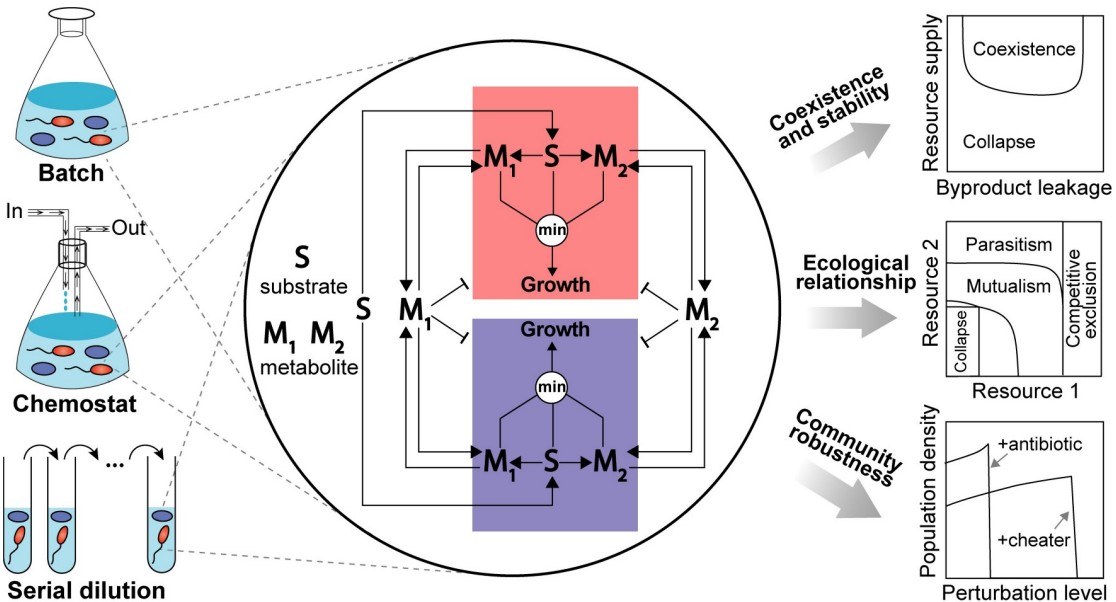

**Fig 1. Schematic diagram illustrating our model and its potential applications in microbial ecology research.** A distinguishing feature of our microbial community model is that each community member harbors a coarse-grained metabolic network. Briefly, the metabolic network transforms growth substrates (**S**) to non-substitutable building block metabolites (**M₁**, **M₂**) and then to biomass whose production rate is set by the supply flux of the most limiting resource among all substrates and metabolites. The intracellularly synthesized metabolites can be secreted to the environment and then utilized by the community as public goods. For simplicity, the network is visually illustrated using one substrate and two metabolites but it can be extended to any number of nutrients. Enabled by the simplified metabolic network, different community members can interact through a variety of mechanisms, including exploitative competitions for shared substrates, cooperative exchanges of nutritional metabolites, and direct inhibition by secreting toxic compounds. Using training data from batch, chemostat or serial dilution cultures, our model can be parameterized to infer microbial processes underlying the data and then used to explore ecological questions and generate testable predictions. Pointed arrows denote the material flow and blunt-end arrows represent growth inhibition.

into building block metabolites, which can be released back to the surrounding environment, used by cells for biomass production, consumed by other non-growth processes, and degraded. Secretable metabolites, when released, can be imported by cells in a way similar to externally supplied substrates, except that their uptake may be inhibited by other substitutable substrates that are assumed to be preferentially used (e.g., catabolite repression). The dynamics of population size change is affected by two elements: population growth and cell death, where the former may depend on both building blocks and substrates. Here the substrate dependency lumps the growth effects from metabolites that are not explicitly modeled, which can substantially reduce model size by defining and choosing model variables for only metabolites known to mediate interpopulation interactions. To model the effects of toxic compounds [27] we allow the growth rate of any cell population can be inhibited by accumulation of toxic metabolites in the environment.

The eight types of reactions can be translated to differential equations. We assumed quasi-steady-state for intracellular substrates and metabolites, as metabolic reactions typically occur at faster time scales compared to ecological dynamics. The time-scale separation thus simplifies our model by excluding intracellular variables, leaving only three types of variables that describe the population density of active cells ($N_l$, $l = 1,2,\cdots,n_c$), the extracellular concentrations of substrates ($[S_i]$, $i = 1,2,\cdots,n_s$), and the concentrations of metabolic byproducts excreted by cells ($[M_j]$, $j = 1,2,\cdots,n_m$). Assuming a chemostat environment with dilution rate $D$ (which reduces to a batch culture when $D = 0$), the differential equations associated with the three state variables are given below (S1 Text)

$$\frac{d[S_i]}{dt} = D\left(S_{0,i} - [S_i]\right) - \sum_{l=1}^{n_c} J_{l,i}^{upt,S} N_l \tag{1}$$

$$\frac{dN_l}{dt} = N_l\left(J_l^{grow} - J_l^{death} - D\right) \tag{2}$$

$$\frac{d[M_j]}{dt} = D\left(M_{0,j} - [M_j]\right) + \sum_{l=1}^{n_c} (J_{l,j}^{leak,M} - J_{l,j}^{upt,M}) N_l \tag{3}$$

where $S_{0,i}$ and $M_{0,j}$ are the feed medium concentrations of substrate $S_i$ and metabolite $M_j$ respectively. $J_{l,i}^{upt,S}$ and $J_{l,j}^{upt,M}$ represent uptake fluxes of substrates and metabolites respectively, $J_{l,j}^{leak,M}$ are metabolite secretion fluxes, and $J_l^{grow}$ and $J_l^{death}$ stand for per-capita growth and death rates respectively. We used Monod kinetics for resource uptake ($J_{l,i}^{upt,S}$ and $J_{l,j}^{upt,M}$), derived mathematical expressions for metabolite leakage ($J_{l,j}^{leak,M}$) and biomass production ($J_l^{grow}$) using the Liebig's Law of the Minimum [28] (growth rate is proportional to the flux of the scarcest resource), and modelled cell death using first-order kinetics with constant specific mortality rate ($J_l^{death}$). The functional forms of these kinetic laws and other details of model derivation are described in S1 Text.

## Example 1: unilateral acetate-mediated cross-feeding

We first applied our modeling framework to a well-documented unilateral acetate-mediated cross-feeding polymorphism evolved from a single ancestral lineage of *E. coli* in laboratory conditions [20] (S1 Text). The community contains two polymorphic subpopulations (*E. coli* subspecies) whose metabolism differs in their quantitative ability to uptake and efflux carbon sources: a glucose specialist strain (CV103) which has a faster glucose uptake rate but cannot grow on acetate, and an acetate specialist strain (CV101) which can grow on acetate but has a

lower glucose uptake rate. CV103 secretes acetate—a major by-product of its aerobic metabolism—and this way creates a new ecological niche for CV101. For simplicity, we assumed that glucose and acetate are fully substitutable resources since *E. coli* cells can grow on either carbon source with similar yields (S1 Text). Compared to its complete form (S1 Fig), the simplified model diverts all glucose flux to acetate that acts as the only growth limiting factor (Fig 2A). Using parameters estimated by manual fitting (Materials and methods, S1 Table), we show that the model accurately reproduced the observed changes in growth and acetate concentration in both monoculture and coculture experiments over time (Fig 2B–2E). Particularly, Fig 2D shows that acetate is toxic to both strains and CV101 is more susceptible. Although Fig 2E shows coexistence of CV101 and CV103 within 40 generations, our model predicts that CV103 would be eventually excluded from the community in the long run (S2 Fig).

The simplified model has 11 parameters, including 8 free parameters, 2 parameters fixed at literature values, and 1 biological constant (S1 Table). To assess parameter uncertainty, we

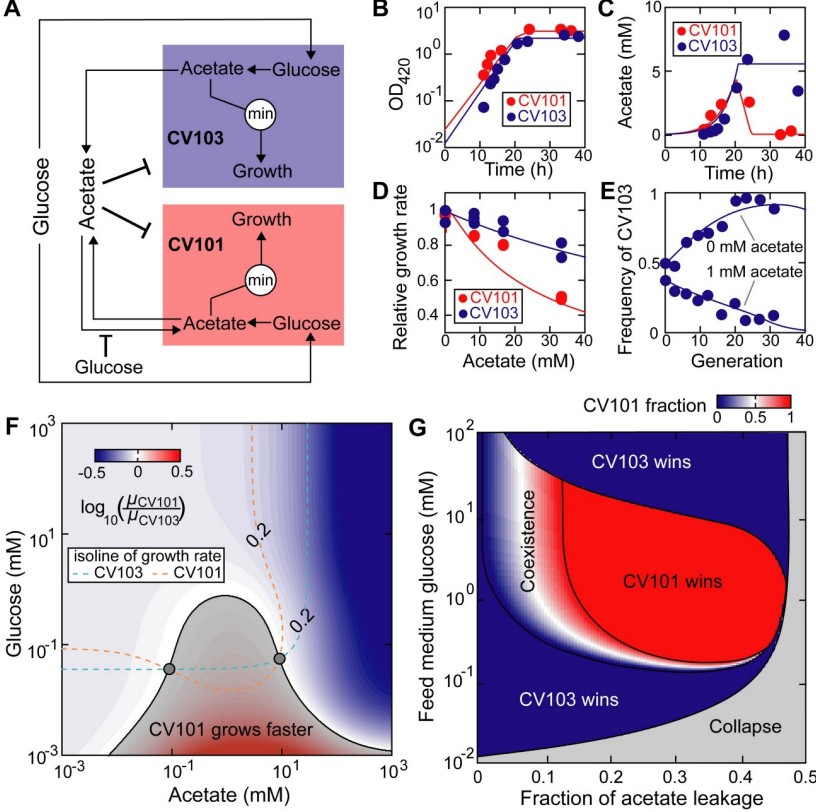

**Fig 2. Unilateral acetate-mediated cross-feeding between evolved *E. coli* isolates.** (A) Schematic diagram of the model. The glucose specialist (CV103) and acetate specialist (CV101) are two *E. coli* mutants with different metabolic strategies [20]: the glucose specialist has improved glucose uptake kinetics while the acetate specialist is able to use acetate as an additional carbon source. At high concentration, acetate inhibits growth of both strains and its uptake by the acetate specialist strain is weakly repressed by glucose. Since glucose and acetate are substitutable, all glucose is converted to acetate which serves as the sole limiting factor for cell growth. (B-E) Manual model calibration. Circles: experimental data; lines: simulations. (B,C) 0.1% glucose-limited batch monoculture without supplementing acetate [20]. (D) 0.0125% glucose-limited batch monoculture supplemented with different concentrations of acetate [56]. (E) 0.00625% glucose-limited chemostat (dilution rate: D = 0.2 h⁻¹) coculture with (1 mM) and without acetate supplementation [20]. The time for one generation is defined as log(2)/D. (F) Growth rate ratio of CV101 to CV103 in the nutritional space. The gray shading indicates when CV101 grows faster than CV103 and the gray circles mark when their growth rates are both equal to the dilution rate 0.2 h⁻¹. (G) The simulated steady-state phase diagram.

sampled posterior distribution of all free parameters using Markov-Chain-Monte-Carlo (MCMC) algorithm (Material and methods), finding that their medians coincide well with the default values obtained by manual fitting and used in the simulations (S3 Fig, S1 Table). Compared to other free parameters, $C_{1,g}$ (half maximum inhibitory concentration of glucose for acetate uptake by CV101) and $I_{3,a}$ (half maximum inhibitory concentration of acetate for CV103 growth) have much wider distributions, suggesting the dataset (Fig 2B–2E) used to constrain the model is relatively insensitive to changes in their values. We did not find strong correlations among parameters, except for the maximum glucose uptake rate of CV101 and CV103 ($V_{1,g}$ and $V_{3,g}$ respectively), which has a Pearson correlation coefficient (PCC) 99.6%. Particularly, the distribution of the acetate leakage fraction has a median 36.7% with interquartile range from 29.8% to 44.6%, which is consistent with the manually optimized value 33.0%. This value suggests that both cell types have nearly equal carbon flux values between acetate secretion and glucose uptake, a quantitative relationship that has been observed in a different *E. coli* strain [29]. The high efflux of acetate may be a consequence of adaptive co-evolution and accumulation of degenerative mutations [20].

Our model indicated that the competition outcome depends on the acetate level in the feed medium (Fig 2E): CV103 dominates the community without acetate supplementation while CV101 dominates when 1 mM acetate was supplemented. Fig 2F outlines the region in the nutritional space when CV101 grows faster than (gray shading) and equal to (shading boundary) CV103. The region has a bell shape with the maximum at 0.81 mM glucose and is almost symmetric around 1 mM acetate. The dose-dependent growth effects can be explained by the conflicting role of acetate which is both a source of carbon and a toxic waste. Acetate at low concentration serves as nutrient for CV101 and increases its growth rate. However, too much acetate is toxic and has stronger inhibitory effects on the growth of CV101 compared to CV103 (Fig 2D). The growth advantage of CV101 conferred by an intermediate level of acetate can be negated at high glucose level ($> 0.81$ mM) due to strong carbon catabolite repression resulting in reduced assimilation of acetate by CV101.

## Coexistence of CV101 and CV103

Coexistence of CV101 and CV103 requires that the growth rate of both strains is equal to the dilution rate. The nutritional space has two solutions (Fig 2F, gray circles) that satisfy the criteria at dilution rate of 0.2 h$^{-1}$ (the value used in the experiment [20]). We then constructed a phase diagram (Fig 2G) that spans a wide range of acetate leakage fraction and the feed medium glucose concentration via simulations. Since acetate is not supplemented, increasing glucose supplementation induces higher release of acetate to the environment. The entire phase space is divided into five distinct regions with four outcomes, including population collapse, extinction of CV103 (CV101 wins), extinction of CV101 (CV103 wins) and stable coexistence. In general, CV103 wins when the supplementation level of glucose is either very low (acetate level is too low to compensate for the growth disadvantage of CV101 due to slower glucose uptake) or very high (acetate level is too high to be toxic and strongly inhibits CV101). Stable coexistence can be maintained within a narrow range of acetate leakage fraction. We show that the coexistence region is robust to changes in the two most uncertain parameters determined by MCMC (S4 Fig). Note that the narrow coexistence regime does not necessarily conflict with the observed transient coexistence in Fig 2E because the theoretical phase diagram was constructed at steady state when time goes to infinity.

Using Chesson's coexistence theory [30], the boundaries of the coexistence region can be interpreted as the conditions when the fitness (growth rate) difference between CV101 and

CV103 is exactly balanced by the stabilizing effects of their niche differences (differential use of carbon sources; in general, it is a collective name for all mechanisms that lower interspecific competition relative to intraspecific competition). When acetate is not leaked (i.e., the acetate leakage fraction is 0), there is no niche difference (the only available carbon source is glucose) and the fitness difference is determined by the basal growth advantage of CV103 due to faster glucose uptake rate. Increasing leakage fraction of acetate leads to higher niche difference since acetate accumulation in the culture allows CV101 to utilize acetate as alternative carbon source and effectively reduces inter-population competition with CV103 for glucose. Meanwhile, increased acetate leakage also causes CV101 to grow faster, first reducing the fitness difference between the two strains to 0 (by overcoming its basal growth disadvantage) and increasing the difference afterwards. As the acetate leakage fraction increases, the lines of niche and fitness difference can possibly have two intersection points (S5 Fig), between which CV101 and CV103 coexist stably because their fitness difference is smaller than their niche difference.

## Example 2: bilateral amino-acid-mediated cross-feeding

The second community is characterized by a synthetic cross-feeding mutualism between lysine and leucine auxotrophies of *E. coli* [21] (S1 Text). The two mutants differ only by single gene deletions in the lysine (Δ*lysA*) and leucine (Δ*leuA*) biosynthesis pathways. Neither mutant can grow in monoculture, but their coculture can survive by exporting essential amino acids that are needed by their partners to the extracellular environment and developing a bidirectional, obligate relationship. For simplicity, we assumed (1) leucine or lysine does not limit growth of the strain that synthesizes it *de novo* (i.e., its producing strain); (2) environment leucine or lysine is not assimilated by its producing auxotrophic strain (S1 Text). Using MCMC algorithm to estimate parameter values of the model that does not take these assumptions (S6 Fig), we justified the second assumption by showing that the amino acid uptake rates by their producing strains are 1–2 orders of magnitude lower than the rates by their non-producing strains (S7 Fig), suggesting that the majority of amino acids in the environment are assimilated by their auxotrophies. However, it is important to note that the assumption is specific to nutrient auxotrophies and may not be generalized to non-auxotrophic, wild-type cells. For example, wild-type *E. coli* cells that are able to synthesize all amino acids *de novo* still grow faster when supplemented with additional amino acids. Using parameters obtained through manual fitting (Materials and methods, S2 Table), we show that the simplified model (Fig 3A) remains effective for quantitatively reproducing the growth and nutrient dynamics in both monoculture and coculture conditions (Fig 3B and 3C).

The simplified model has 15 parameters, including 9 free parameters, 4 parameters fixed at literature values, and 2 biological constants (S2 Table). MCMC simulations confirmed that the posterior median of the free parameters and their values obtained from manual fitting are close to each other (S8 Fig, S2 Table), except that we underestimated the mortality rate constant of the leucine auxotroph ($\eta_{\Delta l}$). Relative to other free parameters, the distributions of $K_g$ (half-maximal concentration for glucose uptake), $\eta_{\Delta k}$ (mortality rate constant of the lysine auxotroph), and $\eta_{\Delta l}$ are much wider and span orders of magnitudes, suggesting that they are loosely constrained by experimental data. In addition, strong negative correlations between the maximum uptake rate and yield of the two amino acids (PCC = -86.9% and -65.5% for leucine and lysine respectively) were found. Finally, the engineered interaction between the lysine and leucine auxotrophies is much weaker with only 0.66% (interquartile range [0.52%, 0.85%]) leucine and 1.13% (interquartile range [0.91%, 1.41%]) lysine released back to the environment (their corresponding values obtained through manual fitting are 0.32% and

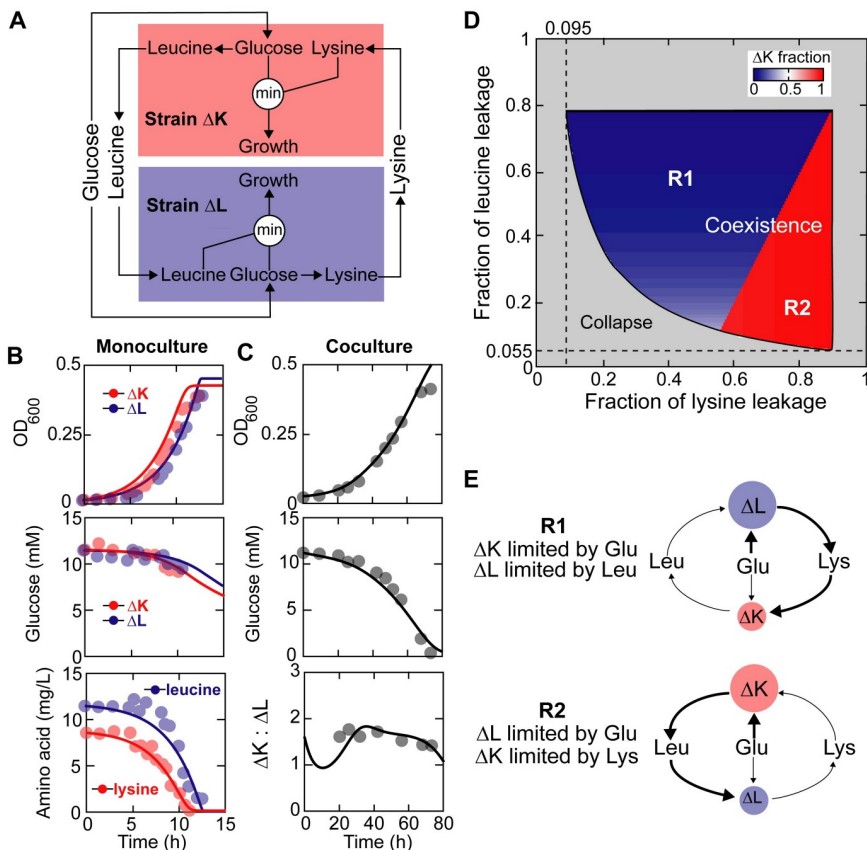

**Fig 3. Bilateral cross-feeding between engineered *E. coli* amino acid auxotrophies.** (A) Schematic diagram of the model. The *E. coli* lysine auxotroph (ΔK) and leucine auxotroph (ΔL) compete for glucose while additionally acquiring essential amino acids from each other. Growth of each auxotroph is determined by the more limiting resource between glucose and the amino acid it needs to grow. (B,C) Manual model calibration. Circles: data; lines: simulation. (B) 2 g/L glucose-limited batch monoculture supplemented with 10 mg/L amino acids [21]. (C) 2 g/L glucose-limited batch coculture without amino acid supplementation [21]. (D) The simulated steady-state phase diagram. The feed medium glucose concentration is 10 mM. (E) The metabolic strategies adopted by ΔK and ΔL in the coexistence regime. All chemostat simulations were run at dilution rate of 0.1 h$^{-1}$.

1.39% respectively), compared to the evolved acetate-mediated cross-feeding interaction (~30% acetate leakage) we studied in Example 1.

## Coexistence of the lysine and leucine auxotrophies

We sought to explore when the two auxotrophic strains coexist in chemostat. Fig 3D shows the phase diagram at different combinations of the lysine and leucine leakage fraction via simulations. We did not see competitive exclusion, which is expected because the interdependence between the two strains is mutually obligate. It is important to note from Fig 3D that the minimum leakage fraction of leucine (5.50%) and lysine (9.50%) required by coexistence at dilution rate 0.1 h$^{-1}$ are far larger than the actual secreted percentages that we fit from experimental data (posterior median 0.66% and 1.13% for leucine and lysine leakage respectively), suggesting that the two engineered strains may not be able to coexist in such condition (but they may coexist at lower dilution rate). Interestingly, the bottom left boundary of the coexistence region describes a negative interaction between the minimum of the two leakage fractions, suggesting

that decreasing leakage of one amino acid must be compensated by increasing leakage of the other in order to satisfy the minimum requirement of coexistence.

Coexistence is possible in the majority of the phase space, suggesting that the community stability is insensitive to the changes in the leakage rates. A striking feature of the diagram is that, increasing the fraction of lysine leakage fraction may trigger a discontinuous, abrupt switch from a steady state dominated by the leucine auxotroph (regime R1) to a steady state dominated by the lysine auxotroph (regime R2). Such abrupt, discontinuous regime shifts are a common feature of microbial communities limited by several essential nutrients [31]. What accompanies with the compositional shift is the qualitative change in the nutrient utilization strategies adopted by the two strains (Fig 3E). Before the switch, growth of the lysine auxotroph is limited by shared glucose while that of the leucine auxotroph is limited by leucine secreted by the lysine auxotroph. When the lysine leakage fraction increases over the threshold of the shift, the lysine auxotroph is limited by lysine secreted by the leucine auxotroph while the leucine auxotroph is limited by shared glucose. Our results indicate that the cellular metabolic strategies that are needed to maintain stable coexistence of the two amino acid auxotrophies vary in a discontinuous manner with continuous changes in amino acid leakage fractions.

## Supplementation of cross-fed metabolites can reverse the sign of microbial social interactions

Cross-feeding interactions within a microbial community may be described as social interactions with costs and benefits to the members involved [32,33]. Those costs and benefits can be altered by environmental perturbations that supply or remove the cross-fed metabolites from the environment. Using the bilateral amino-acid-mediated cross-feeding model, we investigated how the supplementation of amino acids affected ecological relationships between cross-feeders at the steady state (Material and methods).

The phase space that spans a wide range of the leucine and lysine concentrations in the feed medium suggest four possible ecological relationships, including competition, amensalism, mutualism and parasitism (Fig 4A). Mutualism was maintained over a broad range of supplied

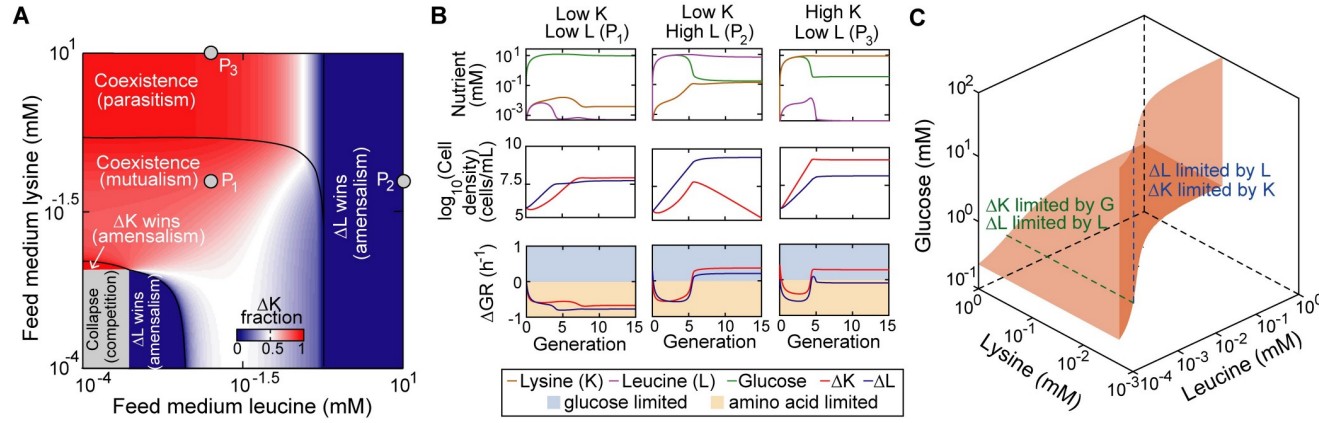

**Fig 4. Impacts of amino acid supplementation on ecological relationships between two amino acid cross-feeders.** (A) The simulated steady-state phase diagram for different levels of amino acid supplementation. (B) Representative system dynamic trajectories of specific phases in (A). ΔGR: the difference between growth rate when set by amino acid as the sole limiting factor and when set by glucose as the sole limiting factor (the minimum of the two determines the actual growth rate). A positive or negative value of ΔGR indicates that cell growth is limited by glucose or amino acid respectively. (C) The isosurface of equal net growth rate (growth rate minus mortality rate) between the lysine and leucine auxotrophies. The dashed lines (blue and green) indicate when their net growth rates are equal to 0.1 h$^{-1}$ (the dilution rate used throughout the figure). Abbreviations: glucose (G); lysine (K); leucine (L); lysine auxotroph (ΔK); leucine auxotroph (ΔL).

amino acid concentrations, even though amino acid supplementation releases the dependence of one auxotroph on the other and is hence detrimental to the mutualistic relationship. In the mutualism regime, glucose is in excess and both amino acid auxotrophies are limited by the essential amino acids they cannot produce (Fig 4B, left column). Further addition of amino acids leads to compositional dominance of one auxotrophic strain, but not necessarily competitive exclusion. Supplementation of leucine destabilizes the community by excluding the lysine auxotroph (Fig 4B, middle column), whereas adding lysine only reduced the relative abundance of the leucine auxotroph, rather than leading to the loss of its entire population (Fig 4B, right column). These results suggest that adding cross-fed nutrients can induce competition between community members that previously interacted mutualistically, and shift positive interactions to negative interactions.

Why supplementation of lysine and leucine cause such asymmetrical long-term effects on the community's composition and stability? We found that the outcome may be dependent on whether one or both auxotrophies is limited by glucose. When glucose limits both auxotrophies (Fig 4B, middle column), competitive exclusion occurs and the leucine auxotroph wins because it has the same glucose uptake kinetics as the lysine auxotroph but lower mortality rate (S2 Table). When only the lysine auxotroph is limited by glucose (Fig 4B, right column), the leucine auxotroph can sustain its population by growing on leucine released by its competitor. Whether coexistence of the two auxotrophies remains stable with increased level of amino acids supplementation can also be understood from the conditions when the net growth rate (growth rate minus mortality rate) of both populations equal to the dilution rate in the nutritional space (Fig 4C). Coexistence requires that the steady state leucine must be equal to $5.25 \times 10^{-4}$ mM, suggesting that supplementing too much leucine would devastate the ability of the system to self-regulate and reach that level at steady state. By contrast, a solution with high lysine concentration is always feasible, which explains why coexistence can be maintained at very high level of lysine supplementation.

## Example 3: multilateral cross-feeding between 14 amino acid auxotrophies

To further demonstrate the utility of our modeling framework, we studied cross-feeding interactions within communities of more than two members. We modeled a community of 14 amino acid auxotrophies engineered from *E. coli* by genetic knockout [22] (Fig 5A). The 14-auxotroph model was directly extended from the 2-auxotroph model developed above by considering each auxotroph can potentially release all other 13 amino acids to the shared environment (S1 Text). Although all feeding possibilities are known, the consumer feeding preferences are not. By fitting experimental data on the population compositions we aimed to infer the unknown feeding pattern—what amino acids and how much they are released by each auxotrophic strain to feed each other.

The model has 269 parameters, including 219 free parameters, 36 parameters fixed at literature values, and 14 biological constants. Parameter values were obtained through both automatic (amino acid leakage fractions) and manual (the rest) data fitting (Material and methods, S3 Table). We show that the fit gave an excellent match to the observed population density fold changes in pairwise cocultures (Fig 5B, PCC = 94%), except for cross-feeding pairs whose fold change values are less than 1. The observed reduction in population density may be caused by cell death in the absence of nutrients but it is difficult to know because the measurement of optical density at low inoculation amount ($10^7$ cells/mL) is highly noisy. For simplicity, we assumed no cell death and set mortality rates to zero in the simulation, which explains why the simulated population density fold changes are always non-decreasing. To compare our model with higher-level models that do not include explicit nutrients, we adopted a Lotka-Volterra

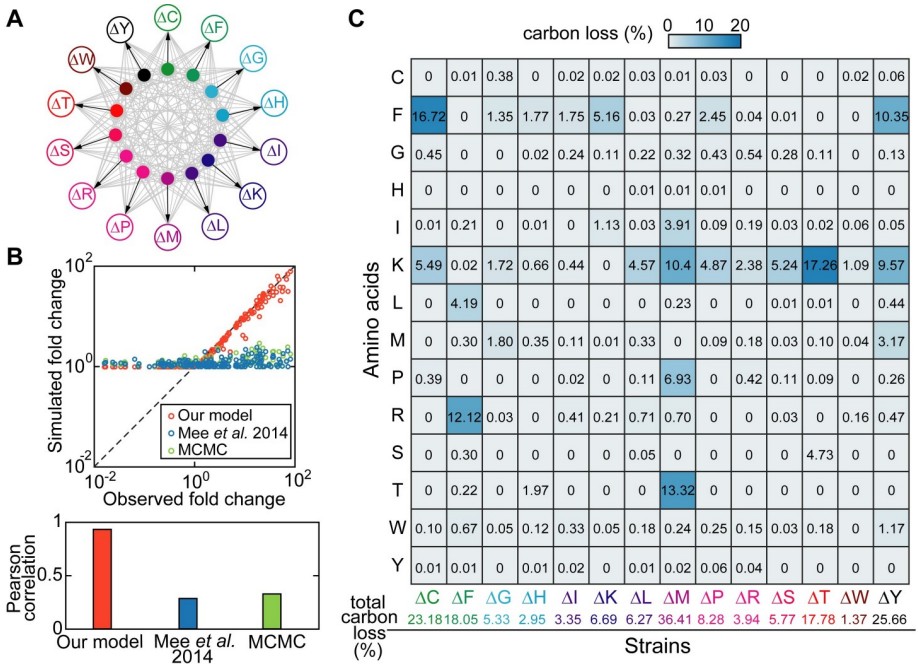

carbon loss (%)

| Amino acids | ΔC | ΔF | ΔG | ΔH | ΔI | ΔK | ΔL | ΔM | ΔP | ΔR | ΔS | ΔT | ΔW | ΔY |
|---|---|---|---|---|---|---|---|---|---|---|---|---|---|---|
| C | 0 | 0.01 | 0.38 | 0 | 0.02 | 0.02 | 0.03 | 0.01 | 0.03 | 0 | 0 | 0 | 0.02 | 0.06 |
| F | 16.72 | 0 | 1.35 | 1.77 | 1.75 | 5.16 | 0.03 | 0.27 | 2.45 | 0.04 | 0.01 | 0 | 0 | 10.35 |
| G | 0.45 | 0 | 0 | 0.02 | 0.24 | 0.11 | 0.22 | 0.32 | 0.43 | 0.54 | 0.28 | 0.11 | 0 | 0.13 |
| H | 0 | 0 | 0 | 0 | 0 | 0 | 0.01 | 0.01 | 0.01 | 0 | 0 | 0 | 0 | 0 |
| I | 0.01 | 0.21 | 0 | 0.01 | 0 | 1.13 | 0.03 | 3.91 | 0.09 | 0.19 | 0.03 | 0.02 | 0.06 | 0.05 |
| K | 5.49 | 0.02 | 1.72 | 0.66 | 0.44 | 0 | 4.57 | 10.4 | 4.87 | 2.38 | 5.24 | 17.26 | 1.09 | 9.57 |
| L | 0 | 4.19 | 0 | 0 | 0 | 0 | 0 | 0.23 | 0 | 0 | 0.01 | 0.01 | 0 | 0.44 |
| M | 0 | 0.30 | 1.80 | 0.35 | 0.11 | 0.01 | 0.33 | 0 | 0.09 | 0.18 | 0.03 | 0.10 | 0.04 | 3.17 |
| P | 0.39 | 0 | 0 | 0 | 0.02 | 0 | 0.11 | 6.93 | 0 | 0.42 | 0.11 | 0.09 | 0 | 0.26 |
| R | 0 | 12.12 | 0.03 | 0 | 0.41 | 0.21 | 0.71 | 0.70 | 0 | 0 | 0.03 | 0 | 0.16 | 0.47 |
| S | 0 | 0.30 | 0 | 0 | 0 | 0 | 0.05 | 0 | 0 | 0 | 0 | 4.73 | 0 | 0 |
| T | 0 | 0.22 | 0 | 1.97 | 0 | 0 | 0 | 13.32 | 0 | 0 | 0 | 0 | 0 | 0 |
| W | 0.10 | 0.67 | 0.05 | 0.12 | 0.33 | 0.05 | 0.18 | 0.24 | 0.25 | 0.15 | 0.03 | 0.18 | 0 | 1.17 |
| Y | 0.01 | 0.01 | 0 | 0.01 | 0.02 | 0 | 0.01 | 0.02 | 0.06 | 0.04 | 0 | 0 | 0 | 0 |
| total carbon loss (%) | 23.18 | 18.05 | 5.33 | 2.95 | 3.35 | 6.69 | 6.27 | 36.41 | 8.28 | 3.94 | 5.77 | 17.78 | 1.37 | 25.66 |

Strains

**Fig 5. Modeling a consortium of 14 amino acid auxotrophies.** (A) Schematic diagram of the model. Each labeled empty circle represents one amino acid auxotroph and each filled circle with the same color corresponds to the amino acid that it is auxotrophic for. Gray arrows indicate production and release of amino acids to the environment and black arrows indicate the uptake of amino acids by their auxotrophies. (B) Scatter plot (upper panel) and Pearson correlation (bottom panel) between observed [22] and predicted cell density fold changes across all pairwise batch coculture of 14 *E. coli* amino acid auxotrophies. Orange circles: our model with manually curated parameters; Blue circles: a Lotka-Volterra-type model with parameters adopted from Mee *et al.* [22]; Green circles: the same Lotka-Volterra-type model with parameters optimized by Markov-Chain-Monte-Carlo (MCMC) algorithm. (C) Predicted amino acid leakage profiles (converted to percentage of carbon loss) for the 14 amino acid auxotrophies. Each value in the matrix describes the fraction of carbon loss due to release of the amino acid in the row by the auxotroph in the column. Abbreviations: cysteine auxotroph (ΔC), phenylalanine auxotroph (ΔF), glycine auxotroph (ΔG), histidine auxotroph (ΔH), isoleucine auxotroph (ΔI), lysine auxotroph (ΔK), leucine auxotroph (ΔL), methionine auxotroph (ΔM), proline auxotroph (ΔP), arginine auxotroph (ΔR), serine auxotroph (ΔS), threonine auxotroph (ΔT), tryptophan auxotroph (ΔW), and tyrosine auxotroph (ΔY).

(LV) type model used in the literature [22,33], which guarantees that cross-feeding is obligate for growth

$$\frac{dx_1}{dt} = C_{1,2}x_2\left(\frac{x_1}{x_1 + b}\right)\left(1 - \frac{x_1 + x_2}{k}\right) \tag{4}$$

$$\frac{dx_2}{dt} = C_{2,1}x_1\left(\frac{x_2}{x_2 + b}\right)\left(1 - \frac{x_1 + x_2}{k}\right) \tag{5}$$

$x_1$ and $x_2$ are cell densities of any two amino acid auxotrophies, $C_{1,2}$ and $C_{2,1}$ are their cooperative coefficients, $b$ is a constant that tunes the saturation concentration of $x_1$ and $x_2$, and $k$ is another constant that represents carrying capacity. We show that the LV-type model can at best achieve a PCC of 33%, using parameters optimized by MCMC algorithm (i.e., parameters from the MCMC sample with the highest PCC). Although this LV-type model has a smaller number of parameters than ours (198 vs. 269), the number of free parameters between the two models is of similar size and comparable (198 vs. 205; note that 14 mortality rates in our model were set to zero).

Fig 5C reports the estimated leakage fractions of 14 amino acids by their amino acid auxotrophies in a matrix form. Although the 14 auxotrophies were derived from the same parent strain, they showed very different profiles of amino acid leakage: some auxotrophies such as the methionine auxotroph ΔM (36.41% total carbon loss) are highly cooperative whereas others such as the tryptophan auxotroph ΔW (1.37% total carbon loss) have very low cooperativity. These differences may be attributed to how metabolic network structure was disrupted to generate the auxotrophies and the concomitant changes in metabolic fluxes. One such example is the strong release (13.32%) of threonine by the methionine auxotroph. Since methionine and threonine biosynthesis pathways branch off from the same precursor homoserine, block of one pathway may lead to increased fluxes of another pathway and leakage of corresponding amino acids. However, the leakage fraction of methionine by the threonine auxotroph is very low (0.1%), suggesting that network topology is not the only factor that affects leakage flux. Since methionine is the most expensive amino acid to produce in terms of ATP consumption [34], its biosynthesis and leakage rates may be tightly regulated and only loosely depend on the level of its precursors.

## The 14-member community converges to a stable coexisting subset at steady state

Besides the pairwise coculture data (Fig 5B), our model also reproduced the population dynamics of serially diluted cocultures of all 14 auxotrophies and four selected 13-auxotroph combinations (Fig 6A). The fit is reasonably good at the log scale, except for the methionine-auxotroph-absent community which seems to undergo non-ecological processes that rescue the threonine auxotroph (ΔT) from the brink of extinction between day 2 and day 3. Quantitatively, the PCCs between observed and predicted values on the log scale are 88.71% (all 14 auxotrophies), 75.30% (lysine-auxotroph-absent), 78.34% (arginine-auxotroph-absent), 52.93% (threonine-auxotroph-absent), and 8.90% (methionine-auxotroph-absent).

As shown in Fig 6A, most amino acid auxotrophies were diluted away very quickly but some, such as the isoleucine auxotroph (ΔI), exhibited transient recovery dynamics after the initial decay. To understand the transient dynamics, we used the same model to infer the concentration dynamics of glucose and all amino acids, which are hidden states (not yet observed) that are relatively costly and inaccurate to measure in experiments. S9 Fig shows that the population density of the isoleucine auxotroph had an initial drop because the isoleucine pool had not been accumulated to a critical size that allows the actual growth to compensate for its mortality and system dilution. As the pool size increases, its net growth rate (growth minus mortality) surpasses the dilution rate and recovers its population density, which eventually levels off when the positive and negative effects are balanced.

By simulating the 14-auxotroph community model to steady state, we further predicted that the initial 14-strain mixture converges to a stable coexisting subset that contains 4 amino acid auxotrophies that are deficient in biosynthesis of isoleucine (ΔI), lysine (ΔK), methionine (ΔM), and threonine (ΔT) (Fig 6B). The predicted coexistence state was successfully validated by two independent observations over 50-day serial dilution [22], a much longer period of time than the duration of the training dataset (7-day serial dilution; Fig 6A). The resource-consumer relationships of the 4-member community are shown in a bipartite network (Fig 7A), where 3 amino acid secretion fluxes were identified as essential (solid arrows) as their deletions resulted in community member loss (S10 Fig). These essential fluxes suggest that the primary feeders for ΔK, ΔM, ΔT are ΔT, ΔI, ΔM respectively; however, none of ΔK, ΔM, ΔT dominates the feeding of ΔI and their contributions to the isoleucine pool in the environment are substitutable.

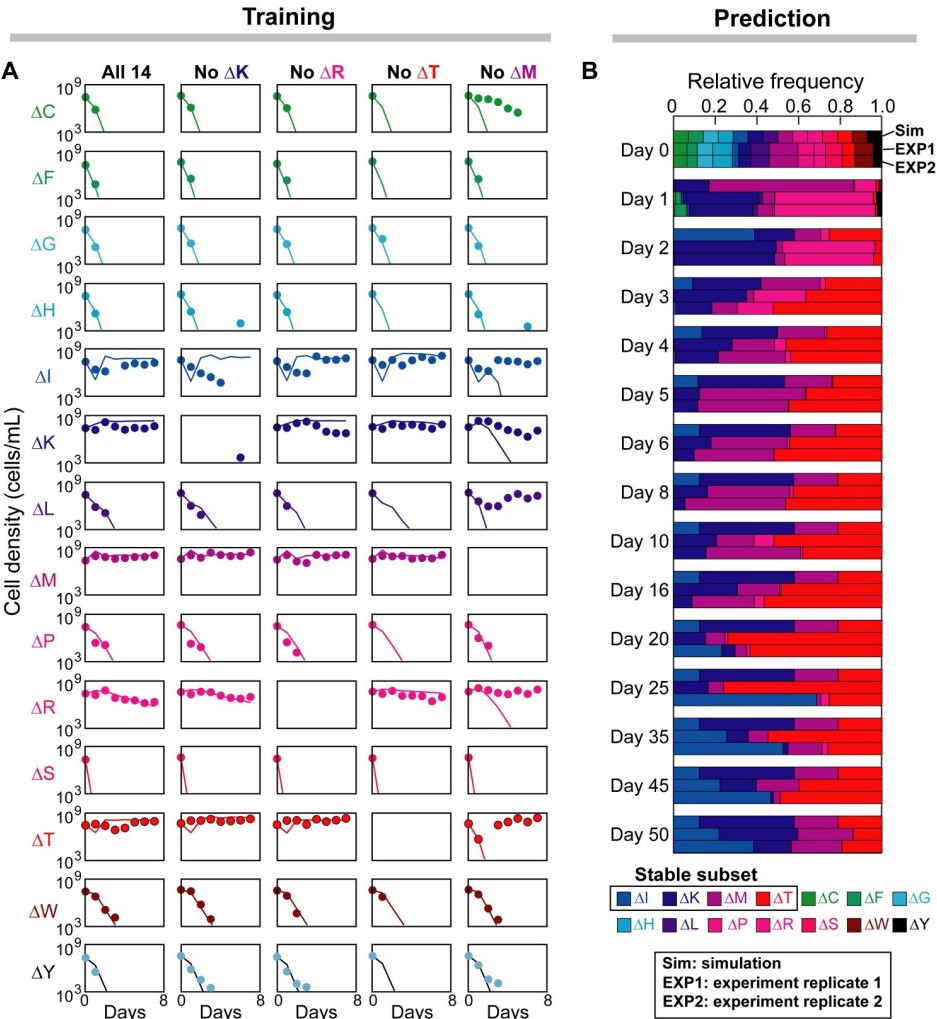

**Fig 6. Prediction of the long-term steady state of the community of 14 amino acid auxotrophies.** (A) Parameters other than the amino acid leakage fractions (obtained from fitting pairwise coculture data in Fig 5) were manually optimized from the observed population density during a 7-day 100-fold serial dilution of one 14-auxotroph and four 13-auxotroph communities. Filled circles: experiments [22]; Lines: simulation. (B) Simulation of the trained 14-member model over 50 daily passages of the community into fresh medium. The observed long-term stable coexistence of a four-auxotroph subset (ΔI, ΔK, ΔM, ΔT) was correctly predicted. The two replicates of experimental observations were adopted from Mee *et al*. [22]. See Fig 5 legend for abbreviations of the names of amino acid auxotrophies.

## Mutualistic cross-feeding network is prone to collapse after external perturbations

Using the model developed above, we computationally tested how external perturbations, including nutrient downshift, the addition of antibiotics, and invasion of cheating phenotypes (the same auxotrophic dependence but no amino acid leakage) affect the stability of coexistence among the 4 auxotrophic strains that would otherwise be stable (Material and methods). The 4-member community was able to cope with these disturbances to a certain extent and remained integrated. Beyond the thresholds, all three perturbation types resulted in community collapse as a result of domino effect (Fig 7B–7D), implying that tightly coupled cooperative communities are fragile and prone to collapse. Since antibiotics inhibit growth of

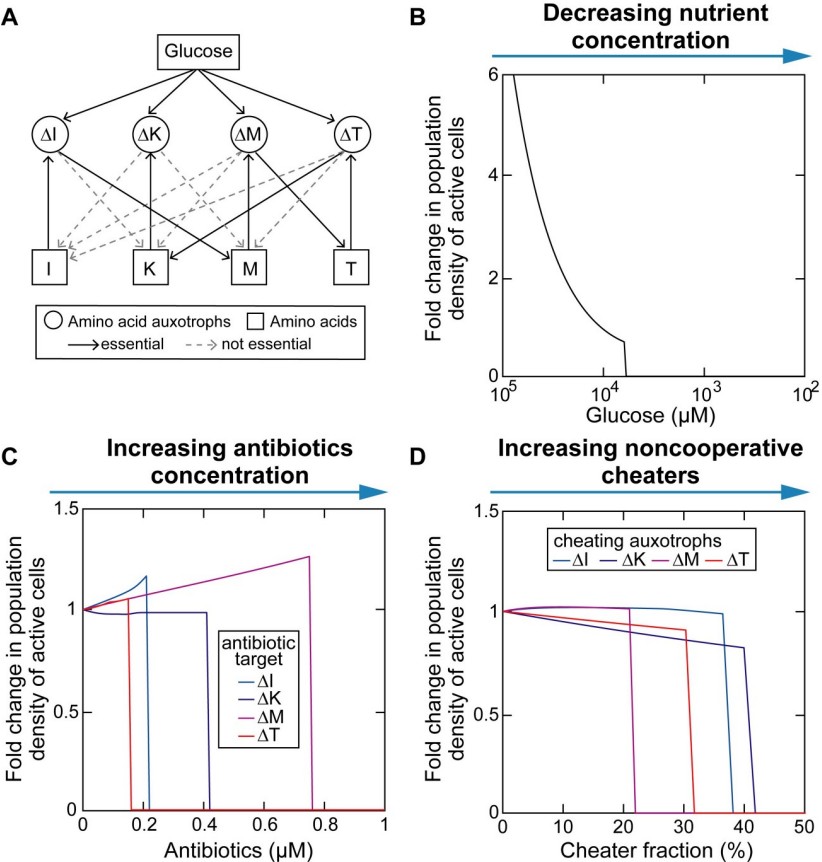

**Fig 7. Collapse of mutualistic cross-feeding network following external perturbations.** 7. (A) Bipartite interaction network of the subset of amino acid auxotrophies that stably coexist over long-term serial dilution (see also Fig 6B). The network contains resource nodes (I, K, M, T for isoleucine, lysine, methionine, and threonine respectively) and consumer nodes (ΔI, ΔK, ΔM, ΔT are their corresponding auxotrophies). Each directed link indicates the presence of a resource-consumer relationship whose corresponding parameter value is not zero. An directed link is essential if its removal leads to loss of community members. (B-D) External perturbations, including decreasing nutrient concentration (B), increasing antibiotic concentration (C), and introducing noncooperative cheaters (D), result in an abrupt collapse of the community when the perturbation level exceeds a certain threshold.

individual strains (targeting consumer nodes in the bipartite network) while cheaters are amino acid sinks (targeting resource nodes in the bipartite network), we identified that ΔT and methionine as the weakest consumer node (Fig 7C) and resource node (Fig 7D) in the bipartite network respectively. Our results suggest that ΔT→K (secretion of lysine by the threonine auxotroph) and M→ΔM (uptake of methionine by the methionine auxotroph)—the outgoing links from the two weakest nodes that are also essential to maintain community integrity—are the weakest metabolic fluxes that may set the resistance level of the community to external perturbations [35].

## Discussion

Predicting population dynamics of a microbial community from interactions between its members is difficult because interaction happens across multiple scales of biological organization [36]. Here we propose a mechanistic ecology model based on a coarse-grained representation of cell metabolism that accurately describes the population dynamics of three laboratory communities with well-defined metabolic exchanges. Previous studies have used genome-scale

models and metabolic flux analysis, but these studies require flux measurements by isotope tracing and metabolomics to fit the adjustable flux parameters. Some success was also achieved by fitting the time series data with coarser-grained ecological models [37–41] such as the gLV equations; however, in gLV-type models, interspecific interactions are phenomenologically defined based on density dependency, which gives little mechanistic understanding of the underlying mechanism [15]. By contrast, our model has explicit formulations of context dependency by representing the chemical flows within and between microbes and thus can explain the metabolic part of microbe-microbe interactions.

When we have limited prior knowledge and data on a given community it becomes critical to choose the right level of details. However, by applying our approach to well-defined laboratory systems, we show that a highly detailed metabolic network is not necessary for developing useful ecological models. In single-bacteria studies, coarse-grained metabolic models have been employed to understand the design principles of metabolic networks and their regulation [42], as well as to predict metabolic flux distributions useful for synthetic biology [43] and industrial [44] applications. Compared to genome-scale models, using coarse-grained models linking ecology and metabolism is simple and has recently become popular [25,45,46]. Depending on the research question, a coarse-grained metabolic network can be created at any level of granularity from a single reaction to the complete whole genome-scale reconstruction. The choice of granularity and how to derive a simpler model from the more complex one are usually empirical but can be facilitated by more systematic approaches to reduce dimensionality.

Our model could extract new insights from those previously published empirical data on well-defined laboratory systems. The analysis shows that unidirectional cross-feeding is equivalent to a commensalism and bidirectional cross-feeding is equivalent to a mutualism. As shown by our study (Figs 2–4) and previous work [27,32], the actual relationship between cross-feeders, however, can be diverse in simple environments (e.g., glucose minimal medium) with constant resource supply due to a combination of positive effects of cross-feeding with negative effects of competition and toxicity of cross-fed metabolites, suggesting that the exact outcome cannot be precisely delineated by the cross-feeding type alone. For example, we predicted that, without supplementation of amino acids, coexistence of the leucine and lysine auxotrophies can only be achieved when one strain is limited in growth by glucose while the other strain is limited by the amino acid it is auxotrophic for (Fig 3E). Although it is theoretically possible that growth of the two auxotrophies is simultaneously limited by the amino acids they are auxotrophic for (i.e., the lysine auxotroph limited by lysine and the leucine auxotroph limited by leucine), this interaction pattern does not occur in the phase diagram because glucose will always be sufficiently depleted to a level that becomes growth limiting to at least one strain. The control of resource pool availability via population dynamics has been demonstrated to be a key mechanism for microbial community to optimize the metabolic strategy of its members to yield resistance to invasions and to achieve maximum biomass [46].

Mechanistic models including explicit nutrients and other realistic features, such as the models presented in this study, can help identify knowledge gaps [47]. For example, recent experiments have demonstrated that the coexistence of two carbon source specialists in the unilateral cross-feeding example is mutualistic in the sense that the consortium is fitter than the individuals [48]. The syntropy can be explained by a null expectation from theoretical ecology models [49]: the glucose specialist provides acetate in an exchange for a service provided by the acetate specialist which scavenges the acetate down to a level at which growth inhibition is insignificant. Although the mechanism of resource-service exchange has been considered in our model, the coexistence regime in the phase diagram (Fig 2G) is competitive, rather than mutualistic. Since mutualism occurs when the reciprocal benefits associated with cross-feeding

outweigh competitive costs [50], our model may predict either or both of lower benefits and higher costs than needed to achieve mutualistic coexistence. Overall, the cost-benefit nature of the cross-feeding interaction between polymorphic *E. coli* strains is more complex than thought and warrants further research.

Our modeling framework explains well the three published experiments but has noteworthy limitations. For example, we assume that the leakage flux is proportional to the conversion rate from substrate to metabolite (proportionality assumption), rather than proportional to the internal metabolite concentration. When does this assumption remain valid and how does it break down? By leveraging our previous experiences in modeling *E. coli* growth and resource allocation [43,51], we developed a coarse-grained single-strain model that explicitly assumes a linear dependency of leakage rate on metabolite concentration (S1 Text). We found that the proportionality assumption remains valid for an internal metabolite when its concentration was perturbed at the upstream, rather than the downstream of the metabolite (S11 Fig). This makes sense because the proportionality assumption couples metabolite leakage with upstream biosynthesis but does not take feedback regulation from downstream reactions and metabolites into accounts. When a perturbation is imposed from the downstream side, the proportionality assumption can lead to undesired behavior such as high leakage flux at low metabolite concentration. Although the assumption remains valid in the context of the current study where resource availability is the only varying external condition, it may prevent us from generalizing our modeling framework to different types of perturbations. Future studies may correct this limitation by incorporating metabolite concentration and associated reaction kinetics.

So far, the current framework has been applied to well-characterized communities with known chemicals and associated interactions which provided a ground through to assess our model. Can the same approach be applied to infer community structure of complex microbiomes (e.g., human gut microbiome) where most of the metabolic exchanges involved in microbe-microbe interactions are still unknown? Our model has the potential if some technical challenges can be solved. First, direct modeling of a real-world microbiome with hundreds of species would be hurdled by too many unknown model parameters. One way to solve this problem is to simply ignore the rare species [38]. Another—arguably better—approach might be by grouping species composition into functional guilds using unsupervised methods that infer those groups from the data alone [52], or to use prior knowledge from genomics or taxonomy to create such functional groups. Second, inferring chemical mediators within a community of interacting populations is a nontrivial task. It can be facilitated by prior knowledge such as searching the literature or leveraging systems biology tools such as community-level metabolic network reconstruction [53]. Finally, our model is nonlinear, so that an efficient and robust nonlinear regression approach for parameter estimation is essential. For a model with similar size to the 14-auxotroph community we analyzed here, non-linear optimization algorithms may fail to converge to a realistic set of parameters and manual parameter selection is often the only feasible approach. Although we primarily chose the manual method to calibrate our models in this proof-of-concept study, manual fitting is a subjective and time-consuming process, requires an expert operator with prior knowledge to choose physically and biologically realistic values, and perhaps more importantly, is unable to infer correlations among parameters. These downsides of manual parameter fitting has, at least for now, precluded it from being applied to large-scale microbial communities. On the positive side, the process of trial-and-error was greatly improved by the speed at which the intermediate-scale model runs simulations on a regular desktop computer. Beyond these technical issues, the model itself can be extended in multiple ways such as incorporating mechanisms of resource allocation [46]. Despite any present limitations, we anticipate that network inference using

mechanism-explicit models can open new avenues for microbiome research towards more quantitative, mechanistic, and predictive science.

## Materials and methods

### Cross-feeding models

The modeling framework presented in this study was developed by integrating a classical ecology model for population and nutrient dynamics with a coarse-grained description of cell metabolism. Custom MATLAB R2018a (The MathWorks, Inc., Natick, MA, USA) codes were developed to perform computational simulations and analyses of all three cross-feeding communities. Please refer to S1 Text for a detailed description of the general modeling framework and its applications to each cross-feeding community.

### Parameter estimation

Our goal was to manually parameterize cross-feeding models directly from experimental data, which are typically cell density and metabolite concentrations in the culture. The manual process of parameter estimation began with initial values of parameters selected to be either equal to their previously reported values or assumed to be of the same order of magnitude based on the literature data. This was followed by the iterative evaluation of model outputs and refinement until sufficient concordance between the model predictions and the experimental data is achieved.

The only exception of parameters that were fit automatically are the amino acid leakage fractions of the 14 amino acid auxotrophies. Under a few assumptions, our model can be simplified and exactly solved for steady state population density in pairwise cocultures (S1 Text). The values of these parameters were then estimated by minimizing the least square error between observed and calculated fold changes of population density across all pairwise batch cocultures. Once obtained, these values were fixed in the process of manually fitting the other parameters of the model.

### Parameter sensitivity analysis

To estimate parameter uncertainty and identify their potential correlations, we used an adaptive MCMC (Markov-Chain-Monte-Carlo) method for sampling the posterior distribution of parameters under constraints of experimental data. We obtained the MATLAB code for this method from https://github.com/mjlaine/mcmcstat. Briefly, this method constructs a sequence of random samples in the parameter space by the Metropolis-Hastings algorithm: at each iteration, the algorithm randomly picks a candidate of the next sample (i.e., parameter set) based on the current sample value. The candidate is accepted with a probability determined by the ratio of the likelihood of the new sample to that of the current sample and the likelihood is given by a negative exponential function where the exponent is the prediction error of our model using a given parameter set. Please refer to the original publication [54] for details of the method.

We ran MCMC simulations for both 2-membebr communities with unilateral and bilateral cross-feeding relationships. The posterior distribution of each parameter was estimated from 100,000 MCMC samples after a burn-in period of 10,000 samples. We assumed a Gaussian prior with standard deviation 0.01. We used symmetric mean absolute percentage error as the cost function that is minimized by the Metropolis-Hastings algorithm:

$$\text{Unilateral cross}-\text{feeding}: \frac{1}{N_{data}}\left(0.1\sum_{i\in Fig.2B,C}\frac{|y_{obs,i}-y_{sim,,i}|}{|y_{obs,i}|+|y_{sim,,i}|}+\sum_{i\in Fig.2D,E}\frac{|y_{obs,i}-y_{sim,,i}|}{|y_{obs,i}|+|y_{sim,,i}|}\right)$$

$$\text{Bilateral cross-feeding} : \frac{1}{N_{data}} \sum_{i \in Fig.3B,C} \frac{|y_{obs,i} - y_{sim,,i}|}{|y_{obs,i}| + |y_{sim,,i}|}$$

where $y_{obs,i}$ is the observed datum $i$, $y_{sim,i}$ is its simulated value, and $N_{data}$ is the total number of data points. Note that the data from different experiments have unequal weights in the unilateral cross-feeding example.

## Simulation of batch, continuous, and serially diluted culture

Deterministic trajectories and their steady states in batch and chemostat conditions were simulated by solving the differential equations from the beginning to the end. Simulations of serial dilution transfer were slightly different in the aspect that the equations were only integrated within each day. The initial condition at the beginning of a day was obtained by dividing all population densities and nutrient concentrations at the end of the previous day by the dilution factor and resetting the feed medium nutrient concentrations to their initial values at day 0.

## Classification of interspecific ecological relationship

We simulated chemostat cocultures of the lysine and leucine auxotrophies at increasing levels of amino acid supplementation in the feed medium, and computed the net effect (+,0,-) of one population on the other by comparing to monoculture simulation. The pairwise ecological relationship between the two populations can then be determined by the signs of their reciprocal impacts [55]: (+,+): mutualism; (-,-): competition; (+,0) and (0,+): commensalism; (-,0) and (0,-): amensalism; (+,-) or (-,+): parasitism; (0,0): no effect.

## Network perturbation

External perturbations were exerted upon the steady state of the 4-auxotroph community. Nutrient downshift was simulated by decreasing the feed medium concentration of glucose at the beginning of simulations. The effects of an antibiotic that inhibit growth of the amino acid auxotroph $i$ was simulated by multiplying the growth rate of the auxotroph by an inhibitory term, i.e., $J_l^{grow} \to J_l^{grow}/(1 + [A]/K_i)$, where $[A]$ is the antibiotic concentration and $K_i$ is the inhibition constant. We assumed antibiotic concentration remains constant and chose $K_i = 1$ $\mu M$. The cheaters of each amino acid auxotroph were simulated by turning off all amino acid leakages of the auxotroph. They were mixed with the resident community in varying ratios at the beginning of simulations. For all three perturbation types, the feed medium glucose concentration is 0.2 wt.% in the unperturbed condition and serial dilution was run to steady state at 60 days.

## Code availability

All computer codes used in this study are available at the following URL: https://github.com/liaochen1988/Source_code_for_cross_feeding_paper.

## Supporting information

**S1 Fig. The schematic diagram of the acetate-mediated cross-feeding model.** For reference, a simplified version is shown in Fig 2A of the main text.
(TIF)

**S2 Fig. The same as Fig 2E of the main text but the simulated curves are shown up to 200 generations.**
(TIF)

**S3 Fig. Posterior distribution of the free parameters of the simplified acetate-mediated cross-feeding model.** (A) Violin plot of the parameter distributions. Gray circles indicate the median of these distributions and red crosses indicate the values obtained through manual fitting and used in simulations. (B) Pairwise scatter plot of these distributions except that the plots along the diagonal are replaced with histograms of parameter values. Parameters not listed here are either fixed to experimentally measured values or biological constants (see S1 Table for their values).
(TIF)

**S4 Fig. Parameter sensitivity analysis of the coexistence region in Fig 2G of the main text.** $C_{1,g}$ and $I_{3,a}$ are the parameters that have the largest uncertainty (S3A Fig). Gray shading indicates the region of stable coexistence. The default values of $C_{1,g}$ and $I_{3,a}$ used to generate Fig 2G of the main text are marked by dashed lines.
(TIF)

**S5 Fig. The schematic diagram showing the qualitative changes in the niche and fitness differences with increasing proportion of acetate leakage.**
(TIF)

**S6 Fig. The schematic diagram of the amino-acid-mediated cross-feeding model.** For reference, a simplified version is shown in Fig 3A of the main text.
(TIF)

**S7 Fig. Violin plot of the posterior distribution of the free parameters of the complete amino-acid-mediated cross-feeding model.** Gray circles indicate the median of these distributions. The red box compares the maximum rates of amino acids uptake by their producing strains ($V_{\Delta l,k}$ and $V_{\Delta k,l}$) and those rates by their non-producing strains ($V_{\Delta k,k}$ and $V_{\Delta l,l}$). Parameters not listed here are either fixed to experimentally measured values or biological constants (see S2 Table for their values).
(TIF)

**S8 Fig. Posterior distribution of the free parameters of the simplified amino-acid-mediated cross-feeding model.** (A) Violin plot of these parameter distributions. Gray circles indicate the median of these distributions and red crosses indicate the values obtained through manual fitting and used in simulations. (B) Pairwise sca_er plot of these distributions except that the plots along the diagonal are replaced with histograms of parameters values. Parameters not listed here are either fixed to experimentally measured values or biological constants (see S2 Table for their values).
(TIF)

**S9 Fig. Inferred dynamics of glucose and 14 amino acids during serial dilution of the 14-auxotroph mixture.** Abbreviations: glucose (Glu), cysteine (C), phenylalanine (F), glycine (G), histidine (H), isoleucine (I), lysine (K), leucine (L), methionine (M), proline (P), arginine (R), serine (S), threonine (T), tryptophan (W), and tyrosine (Y).
(TIF)

**S10 Fig. Essentiality of amino acid secretions in stabilizing the 4-auxotroph community.** A secretion is deemed as essential if its removal leads to strain loss. Each subplot turns off one secretion reaction: $\Delta x \rightarrow z$ indicates the secretion of amino acid z by the amino acid auxotroph

$\Delta x$. Our simulation results suggest that $\Delta I{\rightarrow}M$, $\Delta M{\rightarrow}T$, and $\Delta T{\rightarrow}K$ are essential secretion fluxes. Abbreviations: isoleucine auxotroph ($\Delta I$), lysine auxotroph ($\Delta K$), methionine auxotroph ($\Delta M$), threonine auxotroph ($\Delta T$).
(TIF)

**S11 Fig. A single-strain model of bacterial growth and metabolite overflow.** (A) The schematic diagram. $E_1$ and $E_2$ are the enzymes that control biosynthesis and consumption of the metabolite M respectively. Solid point arrows represent material flow. Dashed point arrows represent positive regulations and dashed blunt arrows represent negative regulations. The gray shading represents a cell. (B-E) Steady state values of various quantities by varying external substrate concentration (the antibiotic concentration is 0 μM). In particular, the model reproduces the observed Monod relationship (circles: [57]) in (B) when the diffusion rate constant ($k_m$, unit: 1/h) is small. (F-I) Steady state values of the same quantities by varying external antibiotic concentration (the substrate concentration is 100 μM).
(TIF)

**S1 Table. Estimated parameter values for the simplified unilateral cross-feeding model.**
(PDF)

**S2 Table. Estimated parameter values for the simplified bilateral cross-feeding model.**
(PDF)

**S3 Table. Estimated parameter values for the multilateral cross-feeding model.**
(PDF)

**S4 Table. Parameter values used in the simulation of the single-strain model.**
(PDF)

**S1 Text. Supporting Information.**
(PDF)

## Acknowledgments

We thank Dr. Michael Mee, Dr. Harris Wang, and Dr. Jennifer Reed for provision and clarification of their experimental data. We also thank Dr. Jinyuan Yan for proofreading early drafts.

## Author Contributions

**Conceptualization:** Chen Liao, Tong Wang, Sergei Maslov, Joao B. Xavier.

**Data curation:** Chen Liao.

**Formal analysis:** Chen Liao, Joao B. Xavier.

**Funding acquisition:** Joao B. Xavier.

**Investigation:** Chen Liao, Tong Wang, Joao B. Xavier.

**Methodology:** Chen Liao.

**Project administration:** Joao B. Xavier.

**Resources:** Joao B. Xavier.

**Supervision:** Joao B. Xavier.

**Validation:** Chen Liao.

**Visualization:** Chen Liao.

**Writing – original draft:** Chen Liao, Tong Wang.

**Writing – review & editing:** Sergei Maslov, Joao B. Xavier.

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
