## [Decision Letter · Decision Letter 0]

3 Apr 2020

Dear Dr. Xavier,

Thank you very much for submitting your manuscript "Modeling microbial cross-feeding at intermediate scale portrays community dynamics and species coexistence" for consideration at PLOS Computational Biology.

As with all papers reviewed by the journal, your manuscript was reviewed by members of the editorial board and by several independent reviewers. In light of the reviews (below this email), we would like to invite the resubmission of a significantly-revised version that takes into account the reviewers' comments.

Both the reviewer expressed interest in the finding of the paper, but also raised a few concerns regarding the model, the presentation of the results, and the parameter fitting.

Also, please note that, as written in the instructions, stating "data available on request from the author" is not sufficient.

We cannot make any decision about publication until we have seen the revised manuscript and your response to the reviewers' comments. Your revised manuscript is also likely to be sent to reviewers for further evaluation.

Sincerely,

Jacopo Grilli

Associate Editor

PLOS Computational Biology

Stefano Allesina

Deputy Editor

PLOS Computational Biology

Both the reviewer expressed interest in the finding of the paper, but also raised a few concerns regarding the model, the presentation of the results, and the parameter fitting.

Also, please note that, as written in the instructions, stating "data available on request from the author" is not sufficient.

Reviewer's Responses to Questions

**Comments to the Authors:**

Reviewer #1: In this manuscript, the authors used resource competition models coupled with abstract cellular metabolism to investigate microbial ecology, focusing on the problem of cross-feeding. The model is of three layers: growth substrates, metabolic intermediates, and biomass. Variants of the model with different assumptions are applied to three experimental E. coli communities: acetate-mediated cross-feeding, bilateral amino-acid-mediated cross-feeding, and multilateral amino-acid-mediated cross-feeding. the experimental data are used to determine the parameters of each model, manually or by combining automatic estimation and manual calibration. Predictions about how external conditions such as resource supply, leakage fraction, antibiotics, and cheaters influence the community behavior are provided by model prediction. Overall, this manuscript is inspiring in successfully connecting a simple theory with experiments. However, I have a number of concerns regarding both the model and the presentation of results.

Major concerns:

(1). Leakage. My largest concern is about the flux of the metabolic intermediates (M in their model). For models of cross-feeding, conversion of intracellular substrate (S^) into intracellular intermediates (M^), then release of some of the intracellular intermediates back into the environment is crucial. However, for simplicity, in the current model no intracellular reactions are explicitly modeled (Equation S7-S9). The extracellular S is directly transformed by cells into the extracellular M, without intracellular metabolic details. Thus in the model there are two divergent pathways, one converting S into M directly in the environment, the other converting S into intracellular M^ that contributes to biomass accumulation. Why does the model assume leakage is proportional to conversion rate from substrate to metabolite, rather than proportional to intracellular metabolite concentration?This biologically unrealistic feature of the model could cause unwanted outcomes. For example, accumulation of M^ inside a cell should presumably increase the leakage of M^ into the environment, but that cannot happen here because there is no such reaction from M^ to M. The current model could lead to strange behavior, e.g. high leakage even at very low internal metabolite concentration, or vice versa. In particular, I would expect that leakage could then depend on growth rate, i.e. a cell growing slowly because of lack of one metabolite could build up another metabolite, and therefore leak the latter at a higher rate. The current model assumes the opposite - slow growth means slow substrate uptake, which means slow leakage. The study would be much more convincing if the model treated leakage more realistically, i.e. as emerging from an internal metabolite pool rather than as a fixed fraction of intake flux. Short of that, the authors need to do much more to justify their approximation, e.g. showing for simple single-strain model that their approximation does a reasonable job compared to a more realistic model over a range of external conditions.

(2) Uptake. Another seemingly unrealistic assumption concerns external amino acid uptake: "Here we assume that each auxotroph only uptakes the amino acid it is unable to produce and does not uptake other amino acids that it can synthesize. (SI p.12)" Certainly, E. coli cells in rich media that includes amino acids grow faster than in minimal media. The authors either need to justify this assumption, which seems contrary to evidence, or do a more careful job explaining how this assumption influences their results, and when it might be important to treat such uptake of external amino acids by non-auxotrophs more realistically.

(3) Stoichiometry. In the model, the flux of intermediates follows both flux balance and stoichiometry, but flux balance and stoichiometry generally cannot be satisfied simultaneously without metabolic overflow. In the model, the flux from the environmental substrate S splits into three fluxes: F1: this flux is converted into environmental metabolites M; F2: this flux is converted into cellular metabolites M^ (then into biomass), and F3: whatever flux is left goes directly into biomass. However, if the growth rate takes the min form of the S and M fluxes, which implies that biomass requires a fixed stoichiometry of S and M, there is flux either from S or M (whichever is not limiting) that cannot enter biomass, and this excess flux need to overflow back into the medium.

(4) Coexistence criteria. The conditions for coexistence are rather underdeveloped. First, in the SI section on coexistence criteria the authors only considered the fixed point solution (setting growth rate of all species equal to the dilution rate), but did not consider the stability of these fixed points. It is unclear whether these analytically derived (approximate) conditions are tested by simulation (and whether the phase diagrams in Figures 3 and 4 are generated from these analytical results or via simulations). Second, more generally the work would benefit from clearer statements about the conditions for coexistence and possible types of coexisting metabolisms. The current model is very similar to that of Taillefumier et al. eLife 2017, which came to definite conclusions about what types of coexistence are possible and when. The current work could benefit from more definite statements along these lines, and a discussion of the relation to this previous study. Here are specific examples that would benefit from more clear statements: on ll. 205-7 "...the glucose specialist releases more acetate than the amount needed to help the acetate specialist overcome its basal growth disadvantage, causing a declining self-balancing capacity of population dynamics and reduced likelihood of coexistence." It's not clear what "likelihood" implies here. Similarly, on ll. 216-21 "Interestingly, growth of the dominant and rare auxotrophs are always limited by its auxotrophic amino acid and glucose respectively, which suggests an implicit negative feedback loop that maintains their relative abundance ratio before and after the switch: increasing population size of the dominant auxotroph impairs the growth of the rare auxotroph by consuming more glucose but eventually, its own growth is inhibited because a smaller amount of amino acid it needs to grow can be produced by its partner." This phrasing is rather complex, and to my mind ambiguous about the origins of the observed behavior. One suggestion is for the authors to produce a graphical representation of the conditions for coexistence, e.g. showing contours of growth rate = dilution rate for each species in the space of nutrient concentrations, and considering possible points where these could overlap to yield coexistence. Such a graphical approach is standard in the field and can provide clear intuition for the classes of outcomes possible.

(5) Comparison to Lotka-Volterra (LV). Does the model actually perform better than an LV model? One of the central claims of the paper is that the level of modeling including explicit nutrients is superior to higher-level models. This sounds reasonable, but isn't satisfactorily demonstrated. The Authors should formulate an LV model with comparable complexity and demonstrate that the current model is actually superior. The authors do provide a comparison to the model in Mee et al., but that model is not sufficiently well explained or justified to judge whether this is a fair comparison.

(6) Parameter sensitivity analysis. Since the analytical results require approximations (S34-35), and the parameters are manually tuned, I think a parameter sensitivity analysis for coexistence is quite important.

Minor points:

(7) Why does the coexistence regime in Figure 3 so narrow? In Lenski's expeirment, the coexistence is quite stable.

(8) What is the definition of a generation? Presumably ~ 1/D.

(9) Fig. 3B - why is the leakage rate set so high, at 0.5?

Reviewer #2: In this manuscript, the Authors provide a detailed study of a consumer-resource model augmented by explicit cross-feeding dynamics of metabolites and resource dynamics. The Authors compare the model output to three experiments between cross-feeding E. coli strains at increasing levels of complexity, and discuss the results in the light of ecological interactions and community robustness. The Authors propose that by fitting the community dynamics to their modeling framework one can quantify exchange rates that are difficult to measure experimentally. I have enjoyed reading the manuscript, which I find interesting and well written. I agree with the Authors conclusion, but I have one major concern that I think should be addressed.

In the introduction and in the discussion, the Authors mention the fact that manual fitting of a model has the downside of being subjective, time consuming and requiring an expert operator. I would add that perhaps the main downside of manual fitting is that one cannot infer correlations among parameters. Especially for chemical reaction models, where chemical species are produced and consumed at unknown rates, fitting parameters can be highly correlated because one can often rescale production and consumption rates to achieve the same average species concentrations. Especially given that the Authors propose this as a framework that can be used to quantify exchange rates that are hard to measure experimentally, I think that the paper requires a rigorous parameter identification approach. At least for the two-species communities, it should be feasible to adopt Markov-Chain-Monte-Carlo methods to estimate the posterior distribution of the parameters, and highlight possible correlations among them. This would also allow the Authors to report confidence intervals for various quantities that are inferred from the data.

Minor points:

- In line 128, the Authors mention that they used Monod kinetics for the cell death. My interpretation of this statement is that cell death is a saturating function of some chemical species or of cell density, but this doesn't seem to be the case looking at the equations, where cell death occurs at a constant rate.

- Line 165-167: it would help to state explicitly here that the strains that are auxotroph for lysine and leucine can export them to the extracellular environment

- Supplementary Tables: I would suggest adding a column with literature values for those best-fit parameters that can be compared to experimentally-measured values

**Have all data underlying the figures and results presented in the manuscript been provided?**

Reviewer #1: Yes

Reviewer #2: Yes

PLOS authors have the option to publish the peer review history of their article (what does this mean?). If published, this will include your full peer review and any attached files.

Reviewer #1: No

Reviewer #2: No
---

## [Decision Letter · Decision Letter 1]

9 Jul 2020

Dear Dr. Xavier,

We are pleased to inform you that your manuscript 'Modeling microbial cross-feeding at intermediate scale portrays community dynamics and species coexistence' has been provisionally accepted for publication in PLOS Computational Biology.

Best regards,

Jacopo Grilli

Associate Editor

PLOS Computational Biology

Stefano Allesina

Deputy Editor

PLOS Computational Biology

Reviewer's Responses to Questions

**Comments to the Authors:**

Reviewer #1: The authors have addressed all of my concerns in the revised manuscript. I am now happy to recommend publication in PLOS Computational Biology.

Reviewer #2: I think that the Authors did a good job in revising the manuscript. I have only one minor comment left: looking at Fig. 3, it seems that some of the parameters' posterior distributions (C_1,g and I_3,a) appear to have a lower bound imposed on them (both appear to cluster at the lower end of the domain). Was a lower bound imposed on the MCMC search? I haven't seen this stated in the methods.

**Have all data underlying the figures and results presented in the manuscript been provided?**

Reviewer #1: Yes

Reviewer #2: None

PLOS authors have the option to publish the peer review history of their article (what does this mean?). If published, this will include your full peer review and any attached files.

Reviewer #1: No

Reviewer #2: No

---

## [Editor Report · Acceptance letter]

7 Aug 2020

PCOMPBIOL-D-20-00308R1 

Modeling microbial cross-feeding at intermediate scale portrays community dynamics and species coexistence

Dear Dr Xavier,

I am pleased to inform you that your manuscript has been formally accepted for publication in PLOS Computational Biology. Your manuscript is now with our production department and you will be notified of the publication date in due course.

With kind regards,

Matt Lyles
